# Development and Validation of a Primary Care Electronic Health Record Phenotype to Study Migration and Health in the UK

**DOI:** 10.3390/ijerph182413304

**Published:** 2021-12-17

**Authors:** Neha Pathak, Claire X. Zhang, Yamina Boukari, Rachel Burns, Rohini Mathur, Arturo Gonzalez-Izquierdo, Spiros Denaxas, Pam Sonnenberg, Andrew Hayward, Robert W. Aldridge

**Affiliations:** 1Institute of Health Informatics, University College London, 222 Euston Rd., London NW1 2DA, UK; neha.pathak.09@ucl.ac.uk (N.P.); claire.zhang.19@ucl.ac.uk (C.X.Z.); yamina.boukari.19@ucl.ac.uk (Y.B.); r.burns@ucl.ac.uk (R.B.); arturo.gonzalez-izquierdo@ucl.ac.uk (A.G.-I.); s.denaxas@ucl.ac.uk (S.D.); 2Guy’s & St. Thomas’ NHS Foundation Trust, London SE1 9RT, UK; 3Office for Health Improvement and Disparities, Department of Health and Social Care, 39 Victoria Street, London SW1H 0EU, UK; 4Department of Non-Communicable Disease Epidemiology, London School of Hygiene & Tropical Medicine, Keppel Street, London WC1E 7HT, UK; rohini.mathur@lshtm.ac.uk; 5Health Data Research UK, London NW1 2BF, UK; 6Institute for Global Health, University College London, 30 Guilford Street, London WC1N 1EH, UK; p.sonnenberg@ucl.ac.uk; 7Institute of Epidemiology & Health Care, University College London, 1-19 Torrington Place, London WC1E 7HB, UK; a.hayward@ucl.ac.uk

**Keywords:** migration, phenotype, validation, algorithm, primary care, clinical practice research datalink

## Abstract

International migrants comprised 14% of the UK’s population in 2020; however, their health is rarely studied at a population level using primary care electronic health records due to difficulties in their identification. We developed a migration phenotype using country of birth, visa status, non-English main/first language and non-UK-origin codes and applied it to the Clinical Practice Research Datalink (CPRD) GOLD database of 16,071,111 primary care patients between 1997 and 2018. We compared the completeness and representativeness of the identified migrant population to Office for National Statistics (ONS) country-of-birth and 2011 census data by year, age, sex, geographic region of birth and ethnicity. Between 1997 to 2018, 403,768 migrants (2.51% of the CPRD GOLD population) were identified: 178,749 (1.11%) had foreign-country-of-birth or visa -status codes, 216,731 (1.35%) non-English-main/first-language codes, and 8288 (0.05%) non-UK-origin codes. The cohort was similarly distributed versus ONS data by sex and region of birth. Migration recording improved over time and younger migrants were better represented than those aged ≥50. The validated phenotype identified a large migrant cohort for use in migration health research in CPRD GOLD to inform healthcare policy and practice. The under-recording of migration status in earlier years and older ages necessitates cautious interpretation of future studies in these groups.

## 1. Background

In 2020, international migrants comprised 14% of the United Kingdom (UK) population [1]. Conditions prior to, during and after migration expose individuals to a range of health risks, resulting in differences in health outcomes between migrants and non-migrants in the migrant’s country of arrival [2]. In the UK, there are well-established multi-generational minority ethnic communities but a history of ‘hostile’ migration policies [3]. The study of migrant health is therefore needed to complement the study of ethnic inequalities to understand how migration intersects with ethnicity, as well as its effects over and above ethnicity to shape risk factors for health, physical and mental health outcomes, and healthcare access [4].

While migrants’ hospitalisation and mortality outcomes have been studied on a population level using electronic health records (EHRs) [5,6], primary care outcomes are scarcely investigated at this scale, despite often being the first point of contact with the UK health system and a central part of the National Health Service (NHS) strategy for preventive care [7]. Most studies examining primary care outcomes in UK migrants are qualitative or employ quantitative survey methods. When EHRs have been used, primary care registration data could only be linked to disease-specific migrant health datasets such as tuberculosis screening [8]. Linkage of census data has only been attempted in Northern Ireland for prescriptions outcomes [9]. Additionally, three studies, all conducted by Jain et al., identified migration status without the use of data linkages [10,11,12] within the Clinical Practice Research Datalink (CPRD), one of the largest UK primary care EHR resources. Using predominantly country of birth and language codes, they estimated that 1.3% of individuals aged ≥65 years in CPRD could be identified as international migrants [11]. However, with 67.7% of migrants in England aged between 16 and 64 years old at the time of the 2011 census [13], a large proportion of migrants at younger ages were not identified by these studies.

Thus, a valid migration phenotype, which is a transparent reproducible algorithm using clinical terminology codes [14], is needed to determine migration status for individuals of all ages using UK primary care EHRs in order to study a broad range of migration health outcomes. A migration phenotype should determine the migration status of a large number of individuals who use primary care and are representative of the UK migrant population. CPRD with its associated linked datasets is an ideal database to use in the development of this phenotype so that it can be used to study primary, secondary and tertiary healthcare utilisation, mortality and other health outcomes in migrants from European Union (EU) and non-EU countries.

We aimed to develop a migration phenotype for UK NHS primary care EHRs and assess its validity in individuals of all ages by describing completeness of recording of migration status, as well as representativeness compared to the Office for National Statistics (ONS) country of birth and 2011 census statistics.

## 2. Materials and Methods

### 2.1. Study Design

This is a study validating a migration phenotype for a population-based cohort study of migration health in the UK using linked EHRs. A flow diagram describing this validation study is shown in Figure 1. The protocol for this population-based cohort study was published previously [15]. Briefly, the protocol describes a study in which the validity of a migration phenotype will be assessed, and a main study to be completed if the phenotype is found to be valid. The main study involves applying the phenotype to the linked CPRD dataset to describe primary care and hospital-based healthcare utilisation and mortality in migrants compared with non-migrants. The protocol also describes how patient and public involvement provided guidance on the research priorities, which included preventable causes of inpatient admission, sexual and reproductive health conditions/interventions and mental health conditions.

### 2.2. Ethics and Approvals

This study is based in part on data from the Clinical Practice Research Datalink obtained under license from the UK Medicines and Healthcare products Regulatory Agency (MHRA). It was approved by the MHRA Independent Scientific Advisory Committee (ISAC protocol 19_062R) and carried out as part of the CALIBER programme [16]. The data were provided by patients and collected by the NHS as part of their care and support. The interpretation and conclusions contained in this study are those of the authors alone.

### 2.3. Data Resource

We extracted data from the CPRD GOLD January 2019 build, which comprises approximately 16 million individuals from 761 practices covering 3.53% of the UK population [17]. CPRD GOLD contains de-identified data of patients across a network of general practice’s (GPs) across the UK that use Vision^®^ EHR software. This data source is broadly representative of the age, sex and ethnicity demographics of the UK general population [18].

### 2.4. Inclusion Criteria

We included individuals of all ages in CPRD GOLD between 1 January 1997 and 31 December 2018 whose record was of ‘acceptable’ research quality. This means CPRD has verified that individuals and their GP practice were contributing ‘up-to-standard’ data [18]. An individual was included at the latest of 1 January 1997, their current registration date or the date on which their GP practice started contributing up-to-standard data to CPRD GOLD. An individual was excluded at the earliest of 31 December 2018, the date their care was transferred out of a CPRD GOLD practice, the practice’s last data collection date for CPRD, or the individual’s date of death.

### 2.5. Development of the Migration Phenotype

We created the phenotype using a systematic approach previously developed from the CALIBER platform described elsewhere [19]. The phenotype was created in three stages (exploration, development and implementation) with feedback at each stage from a team of clinicians, computer scientists, epidemiologists, public health practitioners, bioinformatics and migration health experts.

We searched for Read V2 terms relating to international migration using the following: *migrant*, *migrat*, *countr*, *asylum*, *refugee*, *visa*, *abroad*, *born in*, *origin*, *illegal*, *language*, with the asterisk representing a wildcard search operator. The initial list of terms was reviewed and refined by two experts in migration health research. Each term was assigned a category (Figure 1) based on the type of term (“visa status indicating migration to the UK”, “main/first language not English”, “country of birth outside of the UK”, “non-UK origin”) and a category based on the certainty of migration status (“definite”, “probable”, “possible”). Each individual was classified once using their highest certainty of migration category.

### 2.6. Outcomes

The following three outcomes were used:Migration phenotype: The total number of terms used in the migration phenotype.Completeness: The percentage of migrants recorded in CPRD for the whole study period, in each year and at the time of the 2011 census.Representativeness: The percentage of migrants in CPRD compared with annual ONS country of birth statistics [1], and the percentage of migrants in CPRD living in England and Wales on the date of the 2011 census (27 March 2011) compared with census data [20].

### 2.7. Bias

Misclassification may lead to differential bias where migration status is more likely to be recorded for individuals experiencing a specific outcome than those who do not. This could lead to a false association between migration and outcomes studied. We assessed bias by comparing the distribution of migrants recorded in CPRD GOLD to ONS population statistics and created categories of migration status to address differences in the level of certainty of classification across codes included in the phenotype.

### 2.8. Tools

Data were supplied by the CALIBER research team in multiple files and imported into R software for cleaning and analysis. All data cleaning and analysis code has been made available as open-source metadata.

### 2.9. Data Analysis

We counted the number of different terms used in the migration phenotype and in each category of terms described in Table 1 (including calculation of percentages where relevant). We compared the list of terms to the Jain et al. study [11]. To assess completeness, we estimated the distribution by producing counts and percentages of migrants across the study period and at the time of the 2011 census by sex, year of birth, World Health Organization (WHO) region of birth, continent of birth, 13 CPRD practice region (classified by CPRD as 10 regions in England, with Scotland, Wales and Northern Ireland as separate regions) and ethnicity (18 category groupings, then further aggregated into the 6 higher-level groups of White British, White Non-British, Mixed, Asian/Asian British, Black/Black British, Other to address small group sizes; Appendix A).

To assess representativeness, we compared the percentage of migrants in CPRD with annual ONS country of birth statistics [1,20] both visually/graphically and using the chi-squared test for proportions. Ratios were calculated of the proportion of migrants in CPRD compared to ONS country of birth statistics in each year between 2004 and 2018 (from 2004 onwards, ONS data are sectioned into periods January–December for a more consistent comparison across years) [11]. We also compared, visually/graphically and using the chi-squared test, the percentage of migrants in CPRD living in England on the date of the 2011 census with 2011 census data on country of birth [13] stratified by sex, age, geographical region of origin, and ethnicity. Ratios were calculated of the proportion of migrants in CPRD compared to ONS census data.

We conducted subgroup analyses based on the certainty of migration status (i.e., definite, probably, possible).

## 3. Results

### 3.1. Migrant Phenotype

Four hundred and thirty-four terms indicating migration to the UK were identified from the Read Version 2 terminology system and are listed in Appendix A. The majority of terms indicated country of birth outside of the UK (51.84%; 225 out of 434 terms) or having a non-English main or first language (42.16%; 183 out of 434 terms). The remaining terms related to visa status indicated migration to the UK (3.46%; 15 out of 434 terms) or a non-UK origin (2.53%; 11 out of 434 terms).

Sixty-seven read codes included by the previously mentioned studies of migration health in CPRD by Jain et al. were excluded as they were largely related to reading other languages [10,11,12]. The expert group discussed that preferred written language may not always correspond to a person’s main/first language, so these terms were excluded from the present migration phenotype. A further 36 language-, country-of-birth- and origin-related terms were included in the present migration phenotype that were not included by Jain et al.

### 3.2. Completeness

Of the patients in CPRD between January 1997 and December 2018, 2.51% (403,768/16,071,111) had at least one term indicating migration to the UK (Figure 2). 467,189 events indicating migration were coded across 403,768 individuals. Moreover, 44.3% of these 403,768 individuals were classified as “definite” migrants, 53.7% as “probable” migrants, and 2.05% as “possible” migrants. The most commonly coded migration-related events indicated a non-English first/main language 56.8%. The least commonly coded event was related to being of non-UK origin (2.73%). The percentage of migrants in CPRD GOLD increased from 0.20% in 1997 to 3.64% in 2018. Appendix A details the number and percentage of individuals in CPRD recorded as migrants annually between 1997 and 2018. At the time of the 2011 census, 2.52% of CPRD GOLD patients in England and Wales had at least one term indicating international migration, and their demographic characteristics are detailed in Appendix A.

Table 1 summarises the distribution of migrants in CPRD GOLD for the demographic factors of sex, year/decade of birth, ethnicity, region of birth, and primary care practice region. Just over half of migrants were female (53.7%) and the median year of birth was 1982 (IQR 1973–1990). The most common ethnicity amongst all migrants was White Non-British (34.3%) followed by Asian/Asian British (26.7%) and Black/Black British (9.2%). 42.4% of migrants in CPRD GOLD were registered with a London practice, and the proportion of patients in a region that were recorded as migrants was also highest in London (7.44%; Appendix A).

Of the 140,423 patients with country of birth codes that aligned with a WHO region of birth, the most common was European Region (12.5%) followed by African Region (5.86%) and Western Pacific Region (4.36%). Of the 140,641 patients with country of birth codes that aligned with ONS Nomis continent of birth codes, the most common was the Middle East and Asia (12.5%) followed by Europe (12.4%) and Africa (5.86%).

Distribution of sex and year of birth was consistent across certainty of migration status categories. However, ethnicity was better recorded in “probable” migrants with only 7.71% of unknown ethnicity compared to 28.8% of “definite” migrants with unknown ethnicity.

### 3.3. Representativeness

The percentage of patients recorded as migrants increased over time in CPRD GOLD by 4.6 times between 2004 (0.79%) and 2018 (3.64%) compared to the 1.6-fold increase in migrants as per ONS data over the same period (8.89% in 2004 to 14.2% in 2018; Figure 3). “Probable” migrants increased faster than the other two certainty categories, the “possible” certainty category remained poorly recorded throughout.

While the percentage of migrants in CPRD GOLD was consistently lower than in ONS country of birth data (*p* < 0.0001), the ratio of the percentage of migrants recorded in CPRD compared ONS increased over time from 0.09 in 2004 to 0.26 in 2018 (Appendix A). Migrants were under-recorded in CPRD compared to ONS 2011 census data in all age bands (Appendix A), with the highest numbers recorded in age band 25–34 years (5.22% in CPRD and 25.2% in ONS) and lowest in the age band 85 years and older (0.64% in CPRD, 7.83% in ONS). Migrants aged 0–15 years were most well-recorded in CPRD (2.1% in CPRD, 5.8% in ONS, ratio = 0.41), while 85 years and older were the most poorly recorded group (0.64% in CPRD, 7.83% in ONS, ratio = 0.08).

Comparing the whole migrant cohort within CPRD GOLD and ONS 2011 census data (Figure 4 and Appendix A), differences are smallest across age bands between 16 and 49 years old, but greatest for the 0–15-year-old band and age bands above 50 years old. The proportion of females is similarly higher than males in both datasets (52.3% in CPRD and 51.6% in ONS).

The CPRD migrant cohort and migrants in the 2011 census are similar by continent of birth (Figure 5). Migrants were mostly born in Europe (34.9% in CPRD and 36.6% in ONS) or the Middle East and Asia (34.8% in CPRD and 34.5% in ONS).

Among the CPRD migrant cohort with known ethnicity, Asian/Asian British ethnicity was more frequently recorded than amongst non-UK born individuals in ONS census data (40.5% in CPRD and 32.6% in ONS) while the White British ethnic group was recorded less frequently (1.70% in CPRD and 12.6% in ONS). White British migrants in the ONS data are likely to reflect those born to British nationals living abroad, or those who identify as White British post-arrival to the UK [21]. The remaining ethnic groups had approximately similar proportions between datasets (Figure 6). A comparison of ethnicity using the more granular 18 group classification (Appendix A) resulted in small numbers, limiting the ability to draw definitive conclusions.

## 4. Discussion

We developed and evaluated a phenotyping algorithm that identified over 400,000 migrants in CPRD GOLD. The vast majority of these were either “definite” migrants (codes indicating visa or a country of birth outside the UK) or “probable” migrants (codes indicating a first or main language that was not English). Migration status was under-recorded in CPRD GOLD compared to ONS data, particularly in individuals over the age of 50 years, but increased over the years to capture a quarter of the expected proportion of migrants by 2018. The distribution of sex and geographic region of birth was similar between migrants in CPRD GOLD and ONS datasets. Ethnicity was well-recorded in migrants in CPRD, however, the Asian/Asian British ethnic group was overrepresented compared to ONS data.

Several explanations may account for the lower number of migrants identified in CPRD compared with ONS data. Firstly, GPs do not routinely record migration-related information in EHRs. Recording may be limited to situations where, for example, an interpreter is needed, or differential health risks in a recent migrant’s country of birth/origin will affect clinical decision making. Secondly, barriers to primary care experienced by migrants, such as language, discrimination, lack of knowledge about services [22], and fear of data sharing for the purposes of immigration enforcement [3], could affect migrants’ ability or willingness to register with an NHS GP practice. This corroborates findings of lower levels of primary-care registration amongst newly-arrived migrants to the UK [8] and undocumented migrants and asylum seekers making up a large proportion of patients attending non-NHS primary care [3]. The under-recording of migrants could thus represent a lower number of migrants registering with primary care services. Thirdly, barriers to primary care access could also result in lower attendance at consultations, thereby limiting the opportunity for a GP to ask questions on country of birth, language, or visa type. If there are more opportunities to code migration status with increasing time (and more appointments attended) since GP registration, migrants represented in CPRD GOLD may be those who have lived in the UK longer. As such, generalisability of the phenotype only extends to migrants who have registered with primary care, and they are less likely to be newly arrived migrants [8].

The improved recording of migration status over time, in younger age groups, and in certain ethnic groups could also be explained by healthcare provider coding behaviours or patient healthcare utilisation patterns. Improvements in coding migrant status over time could reflect the incentivising of GPs to record main/first language terms as part of the Quality Outcomes Framework between 2008–2011 [23]. These codes made up the majority of the migration phenotype, and the rate of increase in recording over time was faster in “probable” migrants (terms related to a non-English main/first language) than “definite” migrants. The better recording of migration in younger age groups may be explained by children having more routine contact with primary care unrelated to disease or illness, such as for childhood immunisations and developmental checks. Healthcare use at older ages related to chronic disease may not be as readily accessed by migrants. Older migrants may have migrated to the UK before EHRs existed or before clinical coding in EHRs was well established, and their migration status may not have been coded retrospectively. As a smaller proportion of older migrants are recorded as migrants in CPRD GOLD, there may be greater bias when studying health outcomes associated with older age groups. The better representation of migrants in the Asian/Asian British ethnic group could reflect a higher rate of consultations in this ethnic group as previously described in CPRD GOLD [24]. However, GPs could also deem migration to be more relevant to patients from an Asian/Asian British ethnic group, for example, due to assumptions made about language proficiency or specific health risks. Interpretation of findings should take this into account when analysing migration and ethnicity data using this phenotype. Potential sources of bias also affect this study, with the main limitation being misclassification of migration status. Migrants make up considerably less of the general population than non-migrants, and as a result, the percentage of migrants misclassified as non-migrants is likely to be low. This means that estimates of outcomes in the non-migrant group would be minimally influenced by misclassification, whereas estimates of the same outcomes in the migrant group may be influenced to a greater extent. This may occur in particular as a result of the inclusion of language terms in the phenotype. It may also result in selection bias in future studies of outcomes where migration status is more likely to be recorded for individuals experiencing the outcome versus those who do not experience it. Furthermore, the representativeness of CPRD GOLD practices serving migrants compared to all UK GP practices is unknown and may have affected the low percentage of migrants in CPRD in regions such as London (7%) where ONS estimates of Londoners born abroad are much higher (35%) [1]. Migrants are also likely to be more mobile than non-migrants within the UK; as CPRD cannot link an individual’s record from multiple CPRD practices, migrants may be more likely than non-migrants to be incorrectly counted as more than one individual within the dataset. Significant variation exists between GP practices in their recording of patient sociodemographic indicators, and a more resource-intensive source of validation, such as a nationwide survey of GP practices, is needed to examine these issues further.

Other limitations of the phenotype include, firstly, the under-identification of older migrants aged 50 years and over. Whilst we identified 0.99% of individuals aged 65 years and over as migrants on the date of the 2011 English census (Appendix A), Jain et al. identified 1.3% on the 1st of January 2013 [12]. The greater percentage of migrants identified by Jain et al. in this age group could be a result of improved recording of migrant status over time, as discussed previously, and also the inclusion of written language codes. The addition of these written language codes could be explored in the further development of phenotype certainty categories. Secondly, language codes also make up the “probable” category of migrants, likely over-identifying migrants from non-English speaking countries and under-identify migrants from English-speaking countries, subsequently underrepresenting economic migrants who have good English proficiency. Thirdly, aggregation of ethnic groups into six higher-level categories to deal with small group sizes in migrants loses granularity when comparing the CPRD migrant population with ONS statistics by ethnicity to assess representativeness.

The involvement of experts in migration health and CPRD to develop the migrant phenotype was a strength of this study. Compared to previous approaches, we included a further 36 relevant diagnosis terms indicating migration to create a more comprehensive phenotype. We categorised terms according to the certainty of migration status, allowing future studies to study migration health with varying degrees of certainty for how accurately the phenotype identifies migrant patients in CPRD GOLD. The specificity of the phenotype can be improved by omitting the “possible” migrant certainty category (defined by non-UK origin, making up only 2.1% of all migrants). As the proportion of migrants recorded in CPRD GOLD has improved over time, studying healthcare outcomes in more recent years may be of more value. The cohort in later years should be compared to the 2021 census as a matter of priority when these data become available.

The availability of a migration phenotype to identify migrants in CPRD GOLD will enable the study of important public health topics, such as primary care utilisation and sexual and reproductive health outcomes, in a large cohort of migrants; thus, contributing essential evidence to the migration health field. It also provides a framework for further phenotyping work to study migrant health in other primary care databases. Results from this and any future phenotyping work can then be used to inform the development and implementation of policies that promote equitable healthcare for international migrants presenting to primary care.

## 5. Conclusions

We used a migration phenotype to identify a large cohort of the UK migrant population and demonstrated the feasibility of using CPRD GOLD to undertake large-scale population-based migration health research in the UK. This will allow researchers and policymakers to use primary care EHRs to monitor health outcomes and healthcare in migrants for evidence-based action. However, migrants were under-recorded in the CPRD GOLD database compared to ONS population estimates, particularly in older age groups who may have been in the country longer. Migrants in CPRD GOLD were largely representative of the UK migrant population in terms of sex and geographical region of birth. Improvements in the recording of migration status in CPRD were also observed over time.

## Figures and Tables

**Figure 1 ijerph-18-13304-f001:**
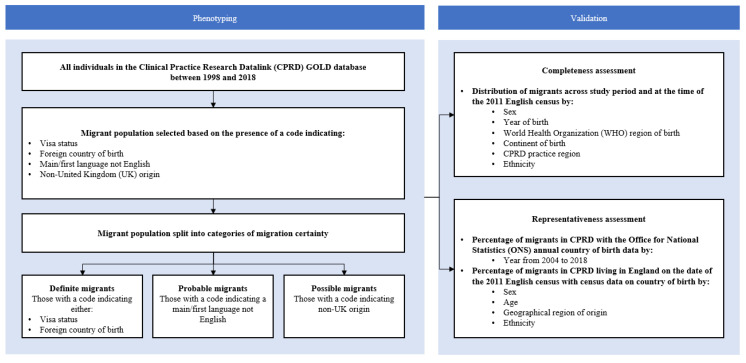
Study flow diagram.

**Figure 2 ijerph-18-13304-f002:**
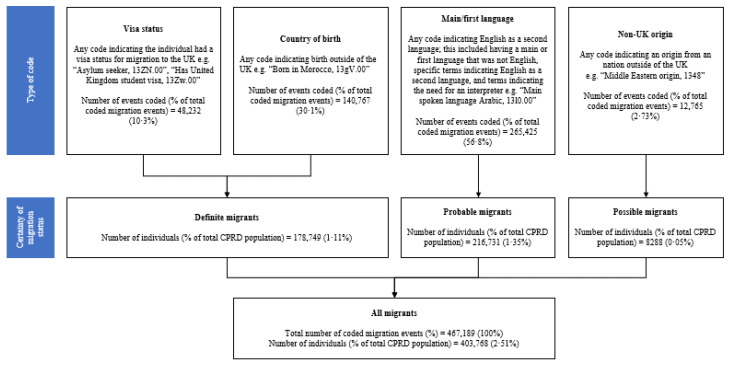
Categorisation of migrants by certainty of migration status using type of migration code.

**Figure 3 ijerph-18-13304-f003:**
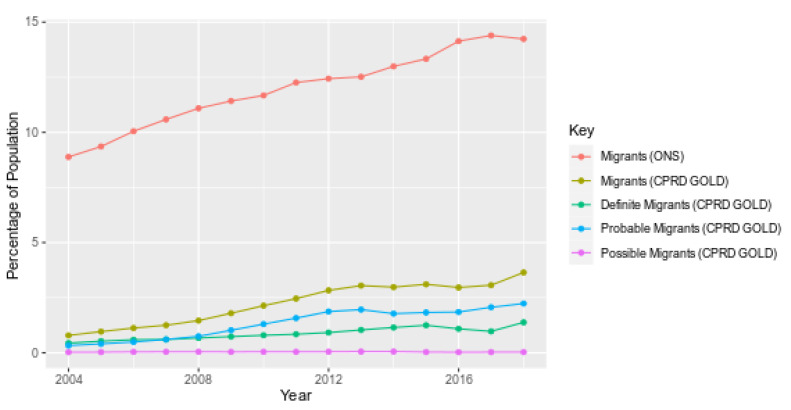
Percentage of international migrants in CPRD and international migrants in ONS by certainty of migration status (2004–2018).

**Figure 4 ijerph-18-13304-f004:**
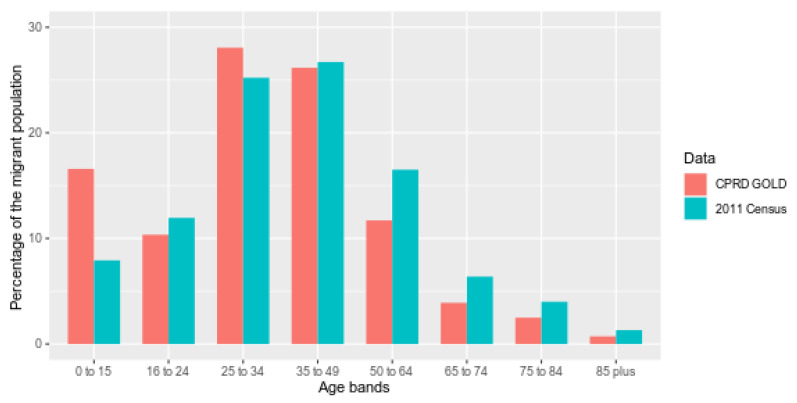
Percentage age breakdown of CPRD (2011) and ONS migrant population at the time of the 2011 census.

**Figure 5 ijerph-18-13304-f005:**
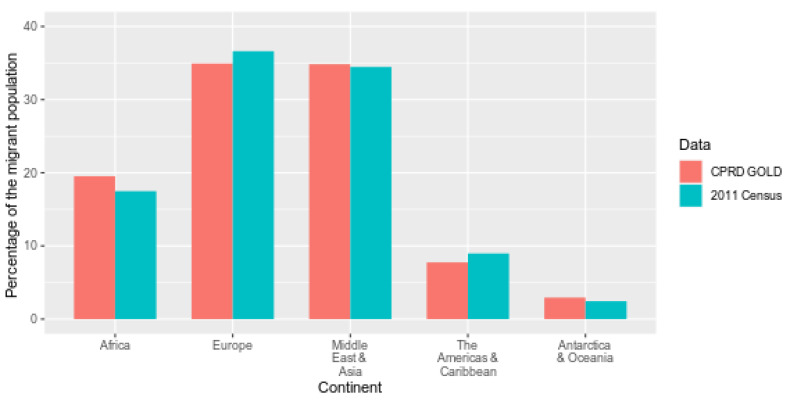
Percentage of migrants in CPRD (2011) and 2011 census according to continent of birth as defined by ONS Nomis.

**Figure 6 ijerph-18-13304-f006:**
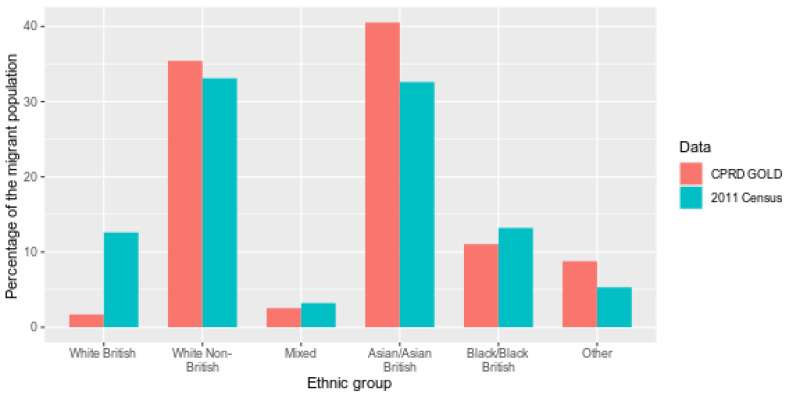
Percentage of migrants in CPRD (2011) and 2011 census by 6 higher-level ethnic groups.

**Table 1 ijerph-18-13304-t001:** Demographic characteristics of recorded migrants in CPRD GOLD by certainty of migration status (1997–2018).

Demographic Characteristic		Migrants (%)	Definite Migrants (%)	Probable Migrants (%)	Possible Migrants (%)	Definite + Probable Migrants (%)
**Totals ***		403,768 (100%)	178,749 (44.3%)	216,731 (53.7%)	8288 (2.05%)	395,480 (97.9%)
**Sex**	Male	187,057 (46.3%)	83,399 (46.7%)	99,849 (46.1%)	3809 (46.0%)	183,248 (46.3%)
	Female	216,704 (53.7%)	95,346 (53.3%)	116,879 (53.9%)	4479 (54.0%)	212,225 (53.7%)
**Year of birth**	1900–1919	456 (0.11%)	194 (0.11%)	212 (0.10%)	50 (0.60%)	406 (0.01%)
	1920–1939	9303 (2.30%)	3387 (1.89%)	5584 (2.58%)	332 (4.01%)	8971 (2.27%)
	1940–1959	31,169 (7.71%)	12,803 (7.16%)	17,292 (7.98%)	1074 (13.0%)	30,095 (7.61%)
	1960–1979	130,715 (32.4%)	62,582 (35.0%)	64,325 (29.7%)	3808 (45.9%)	126,907 (32.1%)
	1980–1999	179,702 (44.5%)	86,459 (48.4%)	90,780 (41.9%)	2463 (29.7%)	177,239 (44.8%)
	2000–2018	52,423 (13.0%)	13,324 (7.45%)	38,538 (17.8%)	561 (6.77%)	51,862 (13.1%)
**Ethnicity**	White British	6125 (1.52%)	3519 (1.97%)	2525 (1.17%)	81 (0.977%)	6044 (1.53%)
	White Non-British	138,410 (34.3%)	48,554 (27.2%)	89,557 (41.3%)	299 (3.61%)	138,111 (34.9%)
	Mixed	11,008 (2.73%)	5373 (3.01%)	5453 (2.52%)	82 (0.989%)	10,826 (2.74%)
	Asian/Asian British	107,630 (26.7%)	35,850 (20.1%)	69,791 (32.2%)	1989 (24.0%)	105,641 (26.7%)
	Black/African/Caribbean/Black British	37,101 (9.19%)	21,100 (11.8%)	14,374 (6.63%)	1627(19.6%)	35,474 (8.99%)
	Other	31,454 (7.79%)	12,819 (7.17%)	18,314 (8.45%)	321 (3.87%)	31,133 (7.87%)
	Unknown	72,040 (17.8%)	51,534 (28.8%)	16,717 (7.71%)	3789 (45.7%)	68,251 (17.3%)
**WHO region of birth ****	African Region	23,675 (5.86%)	23,675 (13.2%)	..	..	..
	European Region	50,588 (12.5%)	50,588 (28.3%)	..	..	..
	Eastern Mediterranean Region	13,701 (3.39%)	13,701 (7.66%)	..	..	..
	Region of the Americas	12,114 (3.00%)	12,114 (6.78%)	..	..	..
	South East Asian Region	14,813 (3.67%)	14,813 (8.29%)	..	..	..
	Western Pacific Region	17,621 (4.36%)	17,621 (9.86%)	..	..	..
	Unknown	263,345 (65.2%)	46,237 (25.9%)	..	..	..
**Continent of birth ****	Africa	23,675 (5.86%)	23,675 (5.86%)	..	..	..
	Europe	50,015 (12.4%)	50,015 (12.4%)	..	..	..
	Middle East and Asia	50,296 (12.5%)	50,296 (12.5%)	..	..	..
	The Americas and Caribbean	12,114 (3.00%)	12,114 (3.00%)	..	..	..
	Antarctica and Oceania	4297 (1.06%)	4297 (1.06%)	..	..	..
	Unknown	263,127 (65.2%)	38,352 (21.5%)	..	..	..
**Practice region**	England	379,844 (94.07%)	163,301 (91.35%)	208,884 (96.38%)	7446 (92.41%)	372,185 (94.11%)
	*London*	171,368 (42.4%)	84,467 (47.3%)	81,530 (37.6%)	5371 (64.8%)	165,997 (42.0%)
	*South Central*	48,740 (12.1%)	26,361 (14.7%)	21,716 (10.0%)	663 (8.00%)	48,077 (12.2%)
	*South East Coast*	43,089 (10.7%)	19,468 (10.9%)	23,324 (10.8%)	297 (3.58%)	42,792 (10.8%)
	*North West*	31,964 (7.92%)	11,666 (6.53%)	20,006 (9.23%)	292 (3.52%)	31,672 (8.01%)
	*West Midlands*	29,629 (7.34%)	5756 (3.22%)	23,556 (10.9%)	317 (3.82%)	29,312 (7.41%)
	*East of England*	24,006 (5.95%)	5394 (3.02%)	18,405 (8.49%)	207 (2.50%)	23,799 (6.02%)
	*South West*	19,734 (4.89%)	8158 (4.56%)	11,463 (5.29%)	113 (1.36%)	19,621 (4.96%)
	*North East*	4980 (1.23%)	611 (0.342%)	4357 (2.01%)	12 (0.145%)	4968 (1.26%)
	*East Midlands*	4594 (1.14%)	1078 (0.603%)	3342 (1.54%)	174 (2.10%)	4420 (1.12%)
	*Yorkshire and The Humber*	1740 (0.43%)	342 (0.191%)	1185 (0.547%)	213 (2.57%)	1527 (0.386%)
	Scotland	12,135 (3.01%)	8090 (4.53%)	3822 (1.76%)	223 (2.69%)	11,912 (3.01%)
	Wales	10,868 (2.69%)	6858 (3.84%)	3618 (1.67%)	392 (4.73%)	10,476 (2.65%)
	Northern Ireland	921 (0.23%)	500 (0.280%)	407 (0.188%)	14 (0.169%)	907 (0.229%)

* Percentages are calculated across columns except for first row ** Country of birth codes only available for those in the ‘definite’ migration certainty category.

## Data Availability

Data used in this study were provided by the Clinical Practice Research.

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
