# Peer review of "Development and Validation of a Primary Care Electronic Health Record Phenotype to Study Migration and Health in the UK"

_ijerph, 2021, doi:10.3390/ijerph182413304_

Round 1

Reviewer 1 Report

I have received the ms titled “Development and validation of a primary care electronic health record phenotype to study migration and health in the UK”. Overall, the paper is well-written and adds thought-provoking evidence on the migrant health burden and its impact on healthcare systems. I have just a few minor comments/suggestions.

Abstract

The abstract seems to exceed the IJERPH word limit.

Methods

Even if the paper relies on a previously published protocols, readers hardly would search it. So, in order to increase the reliability of this paper, authors may want to add summery information on the study protocols. I think that a 100-word paragraph would be ok. Or even less.

Data analysis should be presented at the end of the section; while Bias as study limitations and not in methods. This to enhance readability.

Results

  1. 175.6 is not clear.

Discussion

I’d like to see more on the Public Health impact of the suggested research and tool, as well as its effects on migrant health burden (intended as 10.3390/ijerph17093004)

Reviewer 2 Report

This article aimed to develop a migration phenotype for UK NHS primary care EHRs and assess its validity in individuals of all ages by describing completeness of recording of migration status, as well as representativeness compared to Office for National Statistics (ONS) country of birth and 2011 census statistics. In order to improve the quality of the manuscript, the following clarifications and corrections should be made.

1.Flow chart of selection of the study sample and procedure is suggested.

2 .I recommend describing clearly how to define and validate the selected  diseases.

3.I think that as part of the standard editorial process every manuscript is scanned anti-plagiarisme approach. I have found high similarities between your submission and previous results (www.preprints.org). Please address this and revise the manuscript.

4.Please describe the details of statistics.

5.Please consider the comparison with the other epidemiological studies in other areas using table so make clear the significance of this study.

6.What is the originality and strengths of this study? How physicians or policy makers can deliberate with subjects based on the key findings of this paper?

7.The authors could add the comments related to selection bias in this study to the perceived limitation subsection.

8.Please make sure whether formats of references are described according to the instructions for authors.

If the above suggestions are incorporated and the paper is thoroughly edited, it will be a strong contribution to the literature.

Round 2

Reviewer 2 Report

I am pleased to accept the revised version.

Author Response

Dear Reviewer 2,

We thank you for your feedback and note that a spell check has been carried out. 

Best wishes,

Authorship team